# Comparative Physiological Responses of *Lemna aequinoctialis* and *Spirodela polyrhiza* to Mercury Stress: Implications for Biomonitoring and Phytoremediation

**DOI:** 10.3390/plants14182859

**Published:** 2025-09-13

**Authors:** Chomphoonut Ruamsin, Weerasin Sonjaroon, Sirikorn Khumwan, Arinthip Thamchaipenet, Peerapat Roongsattham

**Affiliations:** 1Department of Genetics, Faculty of Science, Kasetsart University, Bangkok 10900, Thailand; chomphoonut.ru@ku.th (C.R.); skhumwan@mtu.edu (S.K.); arinthip.t@ku.ac.th (A.T.); 2Duckweed Holobiont Resource & Research Center (DHbRC), Kasetsart University, Bangkok 10900, Thailand; weerasin.sonj@ku.th; 3Department of Horticulture, Faculty of Agriculture, Kasetsart University, Bangkok 10900, Thailand; 4Omics Center for Agriculture, Bioresource, Food and Health, Kasetsart University (OmiKU), Bangkok 10900, Thailand

**Keywords:** *Lemna*, *Spirodela*, mercury, stress, phytoremediation

## Abstract

Duckweeds are commonly used as standard ecotoxicological test species and are excellent candidates for phytoremediation due to their adaptability to diverse aquatic conditions. This study investigated the response of two duckweed species, *Lemna aequinoctialis* and *Spirodela polyrhiza*, to mercury-contaminated environments, specifically focusing on their growth rates and photosynthetic performance under mercury stress. Duckweeds were treated with HgCl_2_ at various concentrations (0, 0.1, 0.2, and 0.3 mg/L) in 10% Hoagland’s medium for seven days, after which growth parameters, pigment contents, and chlorophyll fluorescence levels were evaluated. The results showed that increasing mercury concentrations significantly affected growth and photosynthetic pigments in *L. aequinoctialis* and *S. polyrhiza*. Chlorophyll fluorescence analysis indicated that, under higher mercury concentrations, Fv/Fm and Y (II) decreased, while NPQ increased. The findings revealed that *L. aequinoctialis* was more susceptible to mercury toxicity than *S. polyrhiza*. Therefore, *L. aequinoctialis* may serve as a more sensitive species for mercury biomonitoring, whereas *S. polyrhiza* is more resistant and might, for this reason, be more useful for phytoremediation of mercury-contaminated soil.

## 1. Introduction

Mercury pollution is recognized as a global environmental problem with serious implications for Thailand [1]. It poses significant threats to environmental and public health due to its toxic effects on organisms and ecosystems [2]. Once introduced into the food chain, mercury bioaccumulates and circulates, primarily entering the human body through the consumption of contaminated fish, seafood, and wildlife [3]. Mercury exerts widespread toxicological effects on multiple biological systems, including cardiovascular, hematological, pulmonary, renal, immunological, and neurological systems [4]. Chronic mercury exposure has been associated with adverse neurological, renal, and developmental effects in both humans and wildlife [5]. Elevated levels of mercury contamination have been reported in various regions of Thailand, including Khlong Dan, the Mae Tao River near the Thai–Myanmar border, former gold mining sites in the northern provinces, and the Mun River Basin. Dissolved mercury and particulate Hg in water body sites near a gold mine in Pichit Province ranged from 0.58 to 4.19 µg/L, and mercury in sediment ranged from 96.4 to 401.9 µg/kg dry weight. Mercury contamination is elevated in agriculture areas due to fertilizer and pesticide usage. These contaminations affect water, sediments, soil, and aquatic organisms [6,7,8,9,10]. Although Thai authorities have taken steps to monitor and regulate mercury emissions, challenges remain in enforcement, public awareness, and the remediation of contaminated sites, which are further complicated by a lack of research on this issue in Thailand [11].

Currently, several methods are used for heavy metal removal, including chemical precipitation, reverse osmosis, and ion exchange, as well as phytoremediation, which utilizes plants to absorb heavy metals directly from the environment [3,12]. Phytoremediation employs green plants to remediate contaminated media—soil and water—in an economically and environmentally sustainable manner by containing, sequestering, or detoxifying contaminants [13,14]. Several plant species, such as *Lemna minor*, *Salvinia minima*, *Ipomoea aquatica*, and *Centella asiatica*, have been selected for phytoremediation due to their ability to tolerate polluted conditions and remove both organic pollutants and heavy metals [15]. In contrast, pollution-sensitive plants are used as biomonitoring species to detect early-stage contamination in soil and water [16,17].

Duckweed is a fast-growing aquatic plant that adapts well to various aquatic environments. There are only four species reported in Thailand—*Landoltia punctata*, *Lemna aequinoctialis*, *Spirodela polyrhiza*, and *Wolffia globosa* [18]—while *Lemna trisulca* are distributed in Malaysia and Myanmar [19] which are connected to Thailand in the south, north, and west. This situation could imply that duckweed research in Thailand is not well established yet. Duckweed’s small size, simple structure, and rapid growth make it particularly suitable for toxicity testing [20]. Duckweed’s phytoremediation capacity has led to its widespread use in toxicity assessments involving metal-contaminated municipal and industrial wastewaters [21]. Numerous studies have examined the toxic effects of heavy metals on different duckweed species, such as investigations into copper- and mercury-induced oxidative stress and antioxidant responses in *S. polyrhiza* [22], and studies of mercury-induced oxidative stress, DNA damage, and activation of antioxidant systems and Hsp70 induction in *Lemna minor* [23]. Mercury accumulation influences physiological, biochemical, and genomic stability, as well as biomass productivity and chlorophyll content, depending on its chemical form and exposure conditions [24]. Studies specifically focusing on metal toxicity mostly involve *L. gibba*, *L. minor*, and *S. polyrhiza* [25,26,27]; however, research on Thai duckweed species, particularly *L. aequinoctialis*, remains limited.

This study aimed to investigate the growth rates and photosynthetic pigment responses of various duckweed species in mercury-contaminated environments. The objective was to apply these findings to future applications in environmental biomonitoring, bioremediation, wastewater treatment, and mercury absorption.

## 2. Results

### 2.1. Growth Parameters

#### 2.1.1. Frond Area

After treatment with various concentrations of HgCl_2_, the results showed that on day 7, the control group (0 mg/L) had the highest mean frond area of 0.036 cm^2^/frond, while *L. aequinoctialis* exposed to 0.3 mg/L HgCl_2_ exhibited the lowest mean frond area of 0.011 cm^2^/frond (Figure 1A). There was no statistically significant difference (*p* > 0.05) between day 0 and the 0.1 mg/L HgCl_2_ treatment on day 7, indicating growth suppression, although minor chlorosis was observed (Figure 1A and Appendix A). However, treatments with 0.2 and 0.3 mg/L HgCl_2_ resulted in significantly lower frond areas compared to day 0 (*p* < 0.05). The percentage reduction in frond area on day 7 relative to the non-treatment was 2.86%, 22.86%, and 68.57% for the 0.1, 0.2, and 0.3 mg/L HgCl_2_ treatments, respectively (Figure 1A). Visual signs of plant stress and chlorosis supported the observed inhibition of growth (Appendix A).

In *S. polyrhiza*, a decreasing trend in mean frond area was also observed on day 7, although it was not statistically significant (*p* > 0.05) except at the highest concentration (0.3 mg/L HgCl_2_), which showed a significant reduction (*p* < 0.05) (Figure 1B). At this concentration, growth inhibition and notable physiological alterations were evident. The percentage reduction in frond area compared to the non-treatment was 8.82%, 8.82%, and 58.82% for the 0.1, 0.2, and 0.3 mg/L HgCl_2_ treatments, respectively. Frond chlorosis was also observed at higher mercury concentrations (Appendix A).

#### 2.1.2. Relative Growth Rate

The relative growth rate (RGR) of both duckweed species significantly decreased following seven days of mercury treatment. In *L. aequinoctialis*, RGR was reduced by 60.00%, 76.67%, and 86.67% for the 0.1, 0.2, and 0.3 mg/L HgCl_2_ treatments, respectively (Figure 2). A similar decreasing trend was observed in *S. polyrhiza*, with reductions of 48.72%, 66.67%, and 76.92% for the respective concentrations (Figure 2).

#### 2.1.3. Fresh Weight and Dry Weight

The results showed consistent trends, with both the fresh and dry weights/frond number of *L. aequinoctialis* and *S. polyrhiza* decreasing progressively as mercury concentration increased (Figure 3).

#### 2.1.4. Cumulative Colony Breakup

*L. aequinoctialis* showed high sensitivity to mercury stress, with colony breakup observed in all treatments after only 1 h of exposure. At 0.4 and 0.5 mg/L HgCl_2_, nearly all colonies released fronds within 7–8 h, with fronds appearing chlorotic. Higher mercury concentrations accelerated colony breakup (Table 1). In contrast, *S. polyrhiza* displayed a statistically significant increase in frond release only after 24 h at the highest concentration (*p* < 0.01) (Table 2). After mercury exposure, *L. aequinoctialis* was more sensitive to mercury than *S. polyrhiza*, and low concentration led to a longer duration required for colony breakup.

### 2.2. Pigments and Photosynthetic Parameters

#### 2.2.1. Pigment Content

Mercury exposure negatively affected the pigment contents (chlorophyll *a*, chlorophyll *b*, total chlorophyll, carotenoids, and anthocyanins) in *L. aequinoctialis*. At 0.1 mg/L HgCl_2_, no statistically significant reductions were observed in any pigments (*p* > 0.05) (Figure 4A,B). However, beginning at 0.2 and 0.3 mg/L HgCl_2_, significant declines in pigment levels were detected, varying by pigment type. Reductions in pigment content ranged from two- to thirteen-fold (Figure 4A,B). In contrast, the photosynthetic pigments in *S. polyrhiza* were only slightly affected. A non-significant increase in pigment levels was observed at 0.1 mg/L HgCl_2_ (Figure 4C). Anthocyanin remained stable at lower concentration and sharply decreased at higher concentrations. Carotenoid levels gradually declined (Figure 4D).

#### 2.2.2. Chlorophyll Fluorescence

The maximum quantum efficiency of PSII photochemistry (Fv/Fm) in *L. aequinoctialis* was significantly reduced (*p* < 0.05) by day 7 at 0.2 mg/L HgCl_2_. The decline was more pronounced at 0.3 mg/L, where Fv/Fm values could not be detected on days 5 and 7 (Table 3). A similar decreasing trend was observed in *S. polyrhiza*, though the magnitude of decline was less severe (Table 4). Non-photochemical quenching (NPQ) in *L. aequinoctialis* progressively increased with higher mercury concentrations and longer exposure durations (Table 3), indicating elevated stress responses. At high mercury levels, excess light energy could not be used for photochemistry and was dissipated as heat. *S. polyrhiza* also showed increased NPQ, but the response was comparatively less pronounced (Table 4). The effective quantum yield of PSII (Y (II)) in *L. aequinoctialis* declined gradually with increasing mercury concentrations. At 0.2 mg/L HgCl_2_, a significant reduction was observed from day 3 onward, indicating inhibited PSII activity due to mercury stress. At 0.3 mg/L, Y (II) could not be detected from days 3 onward (Table 3). In *S. polyrhiza*, a similar trend was observed, with decreasing Y (II) values as mercury concentration increased (Table 4). However, the degree of inhibition was again less than that observed in *L. aequinoctialis*.

## 3. Discussion

The data demonstrated that the effects of mercury on *L. aequinoctialis* and *S. polyrhiza* were both time- and concentration-dependent. As mercury concentrations increased, duckweed growth consistently declined. These findings align with previous reports on the toxic effects of other heavy metals—such as cadmium, copper, nickel, and zinc—on duckweed, which exhibited visible damage at concentrations of 0.5, 0.5, 4, and 18 mg/L, respectively. Duckweed growth was found to be dependent on the initial metal concentration in the solution, exhibiting a concentration-dependent decline [20,23,28]. Over the 7-day experimental period, increasing mercury concentrations significantly inhibited duckweed growth. Mercury exposure affected various morphological and physiological parameters [29]. Heavy metal presence has been shown to exert toxic effects on *Lemna* fronds by inhibiting soluble proteins and photosynthetic pigments in *Lemna minor* [30], which leads to reductions in surface area, relative frond number (RFN), and both fresh and dry biomass [22,23,29,31,32]. These findings are consistent with the present study, which observed increased frond chlorosis and significant reductions in frond area, RGR, and biomass under mercury stress. Previous studies have reported that mercury caused irreversible damage to *S. polyrhiza* at 0.4 μM (approximately 0.1 mg/L) HgCl_2_ [22]. In this study, however, permanent damage was observed only at concentrations above 0.3 mg/L, possibly due to species- or strain-specific tolerance variations. On day 7, *L. aequinoctialis* exhibited a higher mortality rate than *S. polyrhiza* at the same mercury concentrations, indicating that *S. polyrhiza* is more tolerant to mercury exposure. These findings suggest that *L. aequinoctialis* is more suitable for mercury biomonitoring due to its heightened sensitivity.

Colony breakup was induced more rapidly under higher mercury concentrations in both duckweed species. This phenomenon has been observed in other plant species as well, though the physiological mechanisms vary [33,34,35]. For example, *Athyrium yokoscense*, a fern species, accumulates heavy metals in root tissues and cell walls, preventing their translocation into essential cellular compartments [36]. Likewise, *Amaranthus retroflexus* (redroot pigweed) adapts to heavy metal stress through physiological changes such as reduced photosynthesis and growth [37]. The induced colony breakup observed under mercury exposure in this study likely represents a stress response in duckweed [38]. Similar morphological symptoms, such as decolorization and frond disintegration, have been reported in *L. minor* at mercury concentrations above 0.8 mg/L. At 1.0 and 2.0 mg/L, the fronds completely lost chlorophyll and turned white [31]. The adverse effects observed for other heavy metals such as copper, nickel, cadmium, and zinc are consistent with those caused by mercury [28]. Decolorization and frond disconnection are considered early indicators of metal toxicity in duckweed [21]. Mercury has been found to accumulate in the fronds of duckweed, with the amount of accumulation depending on exposure duration and mercury concentration [23]. Mercury’s chemical form affects its absorption ratio. At 100 µg/L, the absorption equilibrium times for different mercury species were 10 min for inorganic mercury, 20 min for methylmercury, and 40 min for ethylmercury [39]. In heavy metal-contaminated environments, colony disintegration may play a role in plant survival by limiting the transfer of heavy metals from mother fronds to daughter fronds. The release of daughter fronds from metal-stressed mother fronds may enhance their survival potential [40,41,42].

Ethylene is known to regulate plant growth, development, fruit and leaf abscission, and senescence [43,44,45,46,47], and its production increases significantly in response to various stress conditions [48,49,50,51]. Therefore, mercury-induced colony breakup may be driven by stress-induced ethylene synthesis [40,52]. The abscission zone heavily accumulates intra- and intercellular matrix, which is pectin. During the cell separation process, polygalacturonases are induced to degrade pectin at the abscission zone. The matrix has clearly disappeared after cell separation [41,47]. In higher plants, several mechanisms have been proposed for resistance to heavy metal stress, including metal immobilization [53,54]. In this study, *L. aequinoctialis* released fronds as early as 1 h after exposure to 0.5 mg/L HgCl_2_, while *S. polyrhiza* exhibited frond release at approximately 24 h at the same concentration. This differs from a previous report on *L. aequinoctialis* (using the former name, *L. paucicostata* Hegelm), which exhibited frond release at 8 h under 2.0 μmol/L (approximately 0.54 mg/L) HgCl_2_ [40]. Such discrepancies may reflect species-specific differences in sensitivity. When comparing colony breakup responses across heavy metals in *L. aequinoctialis* (using the former name, *L. paucicostata* Hegelm), the order of toxicity based on increasing concentration was: copper (0.1 μmol/L; 0.0064 mg/L) < cadmium (0.8 μmol/L; 0.09 mg/L) < nickel (5 μmol/L; 0.29 mg/L) < zinc (10 μmol/L; 0.65 mg/L) < chromium (40 μmol/L; 2.08 mg/L) [40]. In this study, mercury-induced responses were already evident at 0.3 mg/L, confirming the high sensitivity of duckweed to mercury and other heavy metals at relatively low concentrations. It is important to note that interspecies variation among duckweed species may introduce experimental limitations and differential physiological responses. Overall, *L. aequinoctialis* was more sensitive to mercury than *S. polyrhiza*, and lower mercury concentrations led to a longer time required for colony breakup.

Modifications in photosynthesis often reflect the early symptoms of environmental stress, as photosynthesis functions as a key interface between internal plant metabolism and external environmental conditions. Thus, it is a critical component of overall plant biosynthesis [55,56]. In the present study, the chlorophyll content of *L. aequinoctialis* and *S. polyrhiza* declined under increasing mercury concentrations (Figure 4), although a slight, non-significant induction of chlorophyll was observed in *S. polyrhiza* at 0.1 mg/L HgCl_2_. This response may result from a combination of photoprotective and antioxidant defense mechanisms [2]. Heavy metals, including mercury, have been reported to strongly inhibit soluble proteins and photosynthetic pigments in *L. minor* and various other plant species [2,57]. The current results support this circumstance, showing a significant decrease in photosynthetic pigments in *L. aequinoctialis* and *S. polyrhiza* following mercury exposure. The observed pigment reductions may result from mercury entering the chloroplasts, where it replaces essential metal ions in pigment-binding sites, induces oxidative stress, and interferes with chloroplast function [57,58]. Mercury ions may also impair pigment biosynthesis by inhibiting the uptake and transport of essential micronutrients such as manganese, zinc, and iron [21,57].

In *L. aequinoctialis*, chlorophyll *a*, chlorophyll *b*, and total chlorophyll content decreased by 75.00%, 76.47%, and 78.57%, respectively. In *S. polyrhiza*, the reductions were 50.00%, 55.00%, and 53.57%, respectively. These results indicate that the degradation of chlorophyll *a* and *b* was accelerated under mercury stress, significantly impairing photosynthetic capacity, as both pigments play central roles in light absorption and energy transfer [57]. Anthocyanins, which function as secondary metabolites and potent antioxidants, help mitigate oxidative damage from reactive oxygen species (ROS) generated during heavy metal stress [58,59]. In this study, anthocyanin levels decreased as mercury concentrations increased. Carotenoids, which also possess photoprotective and antioxidant functions [60,61], showed a similar trend of decline under increasing mercury concentrations. Mercury toxicity disrupts the photosynthetic apparatus, particularly photosystem II (PSII), resulting in reduced carotenoid content, diminished quantum efficiency, and elevated oxidative stress [24,62]. After 7 days of exposure to 0–0.3 mg/L HgCl_2_, both duckweed species exhibited significantly reduced photosynthetic pigment levels. These results are consistent with previous findings [23], which reported mercury-induced declines in chlorophyll and carotenoid content across all exposure durations. Several studies have also shown significant negative correlations between mercury concentration and both pigment content and growth rate in duckweed [2,22]. Notably, at 0.1 mg/L HgCl_2_, both species maintained anthocyanins and carotenoid levels similar to the control. This may suggest that anthocyanins and carotenoids are more stable under low mercury stress and may serve as effective indicators or mediators of plant resistance.

Fv/Fm, Y (II), and NPQ are widely accepted as indicators of plant photosynthetic performance and stress [63,64,65]. In this study, a reduction in Fv/Fm reflected decreased PSII maximum potential efficiency under mercury stress, while Y (II) also declined, indicating reduced photochemical efficiency under light conditions. These trends are consistent with prior findings [25,66]. The observed increase in NPQ in both species suggests enhanced energy dissipation as heat, reflecting the activation of photoprotective mechanisms and reduced photosynthetic efficiency—similar to results previously reported [67]. Comparative analysis showed that *S. polyrhiza* maintained better performance in photosynthesis-related parameters (Fv/Fm, Y (II), and NPQ) than *L. aequinoctialis*, indicating a higher tolerance to mercury exposure. Collectively, the data demonstrate that growth and photosynthetic pigment content in *L. aequinoctialis* and *S. polyrhiza* were adversely affected by increasing mercury concentrations. Growth rates and pigment contents declined with increasing HgCl_2_ concentrations. Mercury phytoremediation ability in duckweeds is due to their rapid growth rate and direct absorption through their roots and surface [68]. They can tolerate the toxicity through chelation and sequestration in vacuoles [69]. Moreover, the activation of antioxidant defense systems—such as superoxide dismutase (SOD), catalase (CAT), and peroxidase (POD)—may contribute to mercury tolerance at moderate concentrations [70,71]. However, at higher concentrations, these defense mechanisms appear to be inhibited. In summary, *L. aequinoctialis* was more sensitive to mercury toxicity than *S. polyrhiza* might be due to different mercury management within cells and antioxidant defense mechanisms. Therefore, *L. aequinoctialis* may be more suitable as a biomonitoring species for mercury contamination, whereas *S. polyrhiza* demonstrated greater mercury tolerance potential for use in the phytoremediation of mercury-contaminated aquatic ecosystems.

## 4. Materials and Methods

Samples of *L. aequinoctialis* and *S. polyrhiza* were collected from natural ponds in Thailand (14°00′47.8″ N, 99°58′12.8″ E). Axenic duckweeds were used for this study. Prior to experimental treatment, fronds were acclimatized in 10% Hoagland’s growth medium with 57 μmol m^−2^ s^−1^ and 25 °C for one week. After acclimatization, healthy and uniformly sized duckweed individuals were exposed to various concentrations of HgCl_2_ (0, 0.1, 0.2, and 0.3 mg/L) in the growth medium. Each concentration was tested in five replicates. Following seven days of mercury exposure, growth parameters and chlorophyll content were measured.

### 4.1. Growth Parameters

Frond area was measured on days 0 and 7 of each treatment using Fiji software (version 2.14.0/1.54f) [72], which quantified area based on color threshold. Non-green areas were excluded from the calculation. Relative frond number (RGR) was determined by counting all visible fronds on days 0 and 7. RGR was calculated as RGR = (ln (Frond number at day 7) − ln (Frond number at day 0))/7 [22,73]. Fresh and dry weights were used to assess growth performance. At the end of the experimental period, fronds were blotted dry with paper towels, and fresh weight was recorded. Dry weight was measured after drying the fronds at 60 °C for 10 days until a constant weight was obtained. The cumulative number of colonies exhibiting frond release was recorded hourly for the first 8 h, with additional observations at 24 and 48 h. Mean values and standard deviations were calculated from three replicates.

### 4.2. Pigment Content

Chlorophyll and carotenoid contents were measured using a modified method based on a previous protocol [29]. Approximately 300 mg of fresh fronds were homogenized in 3.0 mL of 95% ethanol and incubated at 4 °C for 30 min. The homogenate was centrifuged (Thermo Scientific™ Pico™ 21 Microcentrifuge, Thermo Fisher Scientific, Waltham, MA, USA) at 13,500 RCF for 3 min, and the supernatant was used for spectrophotometric (BioSpectrometer fluorescence, Eppendorf, Hamburg, Germany) analysis at 647, 663, and 470 nm.

Anthocyanin content was extracted at the end of the experiment following the Wagner method [30]. Fronds were ground in 1 mL of acidified ethanol (ethanol: HCl 99:1 *v*/*v*). The resulting extracts were then centrifuged at 13,500 RCF for 3 min and the sample was diluted in two different buffer solutions—one at pH 1.0 and the other at pH 4.5. The absorbance of each solution is measured at specific wavelengths (520 nm and 700 nm) using a spectrophotometer [74].

### 4.3. Chlorophyll Fluorescence

Chlorophyll fluorescence was measured using a Photosynthesis Yield Analyzer MINI-PAM-II (Heinz Walz GmbH, Effeltrich, Germany). For the dark-adapted measurement, duckweed samples were placed in 1.5 mL microcentrifuge tubes containing 10% Hoagland’s medium and dark-adapted for 30 min. Minimal fluorescence (Fo) and maximum fluorescence (Fm) were recorded to assess the maximum quantum yield of PSII photochemistry (Fv/Fm), calculated as Fv/Fm = (Fm − Fo)/Fm [75]. For the light-adapted measurement, samples were exposed to ambient light for 30 min to reach a steady photosynthetic state. Actinic light was then applied at 45 μmol m^−2^ s^−1^ for 15 s to stabilize photosynthetic activity, followed by a saturating pulse. Effective quantum yield of PSII (Y (II)) and non-photochemical quenching (NPQ) were calculated as Y (II) = (Fm′ − F′)/Fm′ and NPQ = (Fm − Fm′)/Fm′ [75].

### 4.4. Statistical Analysis

Data were analyzed using one-way analysis of variance (ANOVA). Tukey’s HSD test was used to identify significant differences among treatments. Unless otherwise specified, statistical significance was set at *p* < 0.05. All experimental data are presented as the means of five replicates. For cumulative colony breakup, data from three replicates were analyzed by one-way ANOVA with a significance threshold of *p* < 0.01. Tukey’s HSD test was applied to determine significant differences among treatments [76].

## 5. Conclusions

In summary, mercury exerted significant adverse effects on *L. aequinoctialis* and *S. polyrhiza*, with both species responding similarly but to differing degrees. As mercury concentration increased, a gradual decline was observed in frond area, relative frond number (RGR), chlorophyll content, anthocyanin content, carotenoid content, Fv/Fm, and Y (II). Conversely, NPQ increased with higher mercury concentrations, indicating reduced photochemical efficiency of PSII and enhanced energy dissipation as heat. The findings suggest that *L. aequinoctialis* is highly sensitive to mercury and thus a suitable species for use in biomonitoring, while *S. polyrhiza* demonstrates greater tolerance and potential for application in phytoremediation. Nevertheless, practical implementation of duckweed-based systems remains challenging due to factors such as limited root systems, seasonal variability, and issues related to biomass disposal. Another factor needed to consider is environmental water pH, because neutral to weakly acidic conditions have better mercury absorption than alkaline environments [77]. Further research is needed to evaluate mercury-induced oxidative stress by analyzing malondialdehyde and proline levels, the activities of ROS-scavenging enzymes, the accumulation of mercury in duckweeds, and the expression of stress-related genes. Additionally, exploring mercury responses in other duckweed species would enhance the development of effective wastewater treatment strategies in Thailand.

## Figures and Tables

**Figure 1 plants-14-02859-f001:**
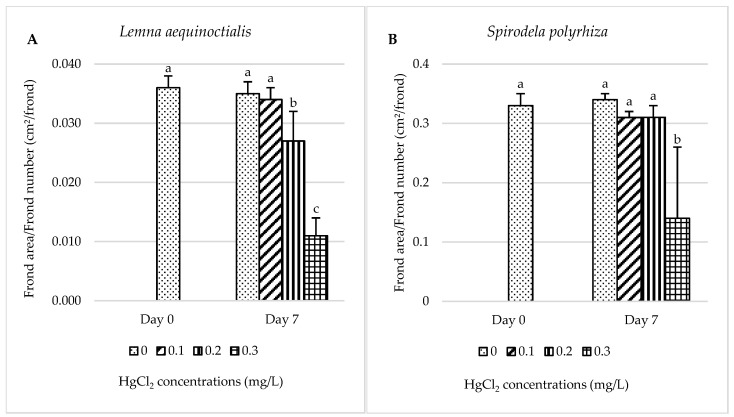
Effects of different concentrations of mercury on the frond area of *L. aequinoctialis* (**A**) and *S. polyrhiza* (**B**) at 0 and 7 days. Same letters indicate no significant differences; different letters indicate significant differences among treatments. Error bars represent standard deviation.

**Figure 2 plants-14-02859-f002:**
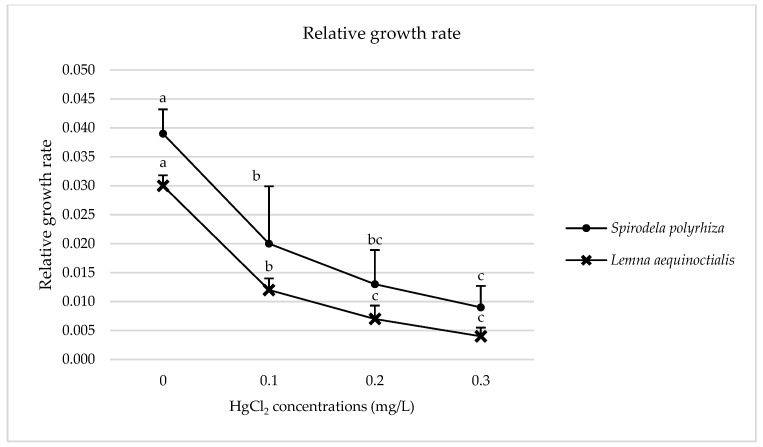
Effects of different concentrations of mercury on the relative growth rate of *L. aequinoctialis* and *S. polyrhiza* on day 7. Same letters indicate no significant differences; different letters indicate significant differences among treatments. Error bars represent standard deviation.

**Figure 3 plants-14-02859-f003:**
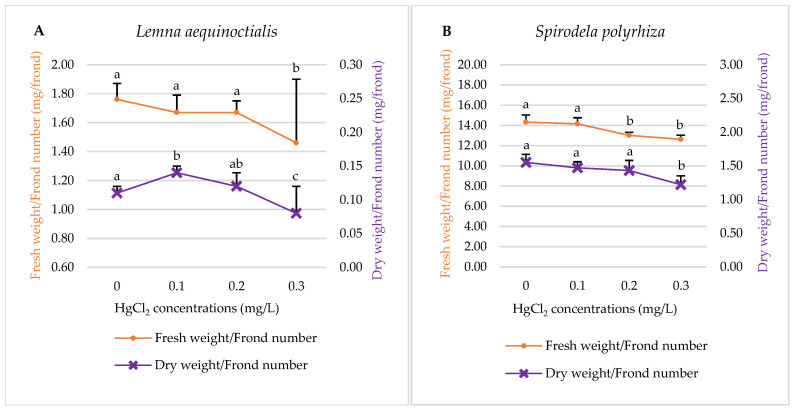
Effects of different concentrations of mercury on the fresh weight and dry weight of *L. aequinoctialis* (**A**) and *S. polyrhiza* (**B**). Same letters indicate no significant differences; different letters indicate significant differences among treatments. Error bars represent standard deviation.

**Figure 4 plants-14-02859-f004:**
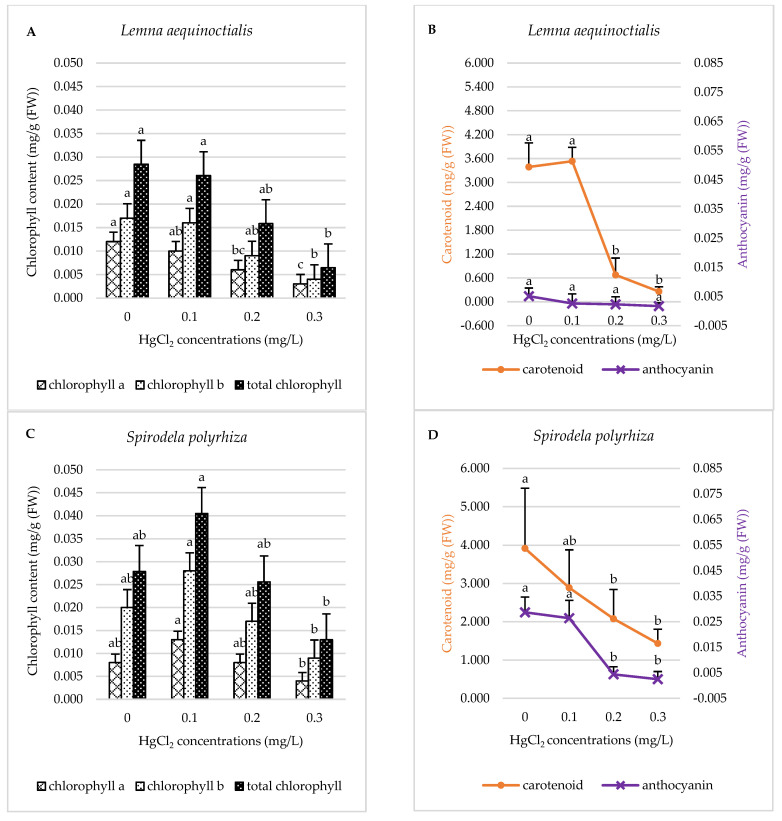
Effects of different concentrations of mercury on the content of chlorophyll *a*, chlorophyll *b*, and total chlorophyll of *L. aequinoctialis* (**A**) and *S. polyrhiza* (**C**). The content of carotenoid and anthocyanin of *L. aequinoctialis* (**B**) and *S. polyrhiza* (**D**). Same letters indicate no significant differences; different letters indicate significant differences among treatments. Error bars represent standard deviation.

**Table 1 plants-14-02859-t001:** Response of *L. aequinoctialis* to mercury versus time and dose. Cumulative increase in colony breakup.

Time (h)	HgCl_2_ (mg/L)					
	0	0.1	0.2	0.3	0.4	0.5
0	0	0	0	0	0	0
1	0	1.00 ± 0.00	0.67 ± 1.20	1.00 ± 1.00	3.67 ± 1.50	5.00 ± 2.00 *
2	0	1.00 ± 0.00	1.00 ± 1.00	1.00 ± 0.00	5.67 ± 1.50 *	8.67 ± 2.10 *
3	0	1.00 ± 0.00	1.00 ± 1.00	2.00 ± 1.00	7.00 ± 1.00 *	9.33 ± 2.30 *
4	0	1.00 ± 0.00	1.00 ± 1.00	5.33 ± 0.60 *	7.67 ± 0.60 *	9.67 ± 2.10 *
5	0	1.00 ± 0.00	1.00 ± 1.00	7.00 ± 0.00 *	9.67 ± 3.10 *	10.67 ± 1.50 *
6	0	1.00 ± 0.00	1.00 ± 1.00	7.67 ± 1.20 *	10.00 ± 2.60 *	10.67 ± 1.50 *
7	0	1.00 ± 0.00	2.67 ± 1.50	8.67 ± 0.60 *	10.33 ± 2.50 *	11.00 ± 1.70*
8	0	1.00 ± 0.00	2.67 ± 1.50	10.00 ± 0.00 *	10.67 ± 2.10 *	11.00 ± 1.70 *
24	0	2.00 ± 0.00	2.67 ± 1.50	10.00 ± 0.00 *	11.00 ± 2.00 *	11.33 ± 1.20 *
48	0	2.00 ± 0.00	2.67 ± 1.50	10.00 ± 0.00 *	11.00 ± 2.00 *	11.33 ± 1.20 *

Note: Shading indicates the shortest time at which a significant difference from the control is observed at *p* = 0.01. Asterisks indicate significant differences from the control at *p* = 0.01.

**Table 2 plants-14-02859-t002:** Response of *S. polyrhiza* to mercury over time and dose. Cumulative increase in colony breakup.

Time (h)	HgCl_2_ (mg/L)					
	0	0.1	0.2	0.3	0.4	0.5
0	0	0	0	0	0	0
1	0	0	0	0	0	0
2	0	0	0	0	0	0
3	0	0	0	0	0	0
4	0	0	0	0	0	0
5	0	0	0	0	0	0
6	0	0	0	0	0	0
7	0	0	0	0	0	1.00 ± 1.00
8	0	0	0	0	0.33 ± 0.60	1.33 ± 1.50
24	0	0.33 ± 0.60	0.67 ± 1.20	1.00 ± 1.00	3.00 ± 1.00	4.00 ± 1.00 *
48	0	0.67 ± 0.60	1.33 ± 1.50	2.00 ± 1.00	3.33 ± 1.50	4.00 ± 1.00 *

Note: Shading indicates the shortest time at which a significant difference from the control is observed at *p* = 0.01. Asterisks indicate significant differences from the control at *p* = 0.01.

**Table 3 plants-14-02859-t003:** Effects of different concentrations of mercury on the maximum quantum efficiency of PSII photochemistry (Fv/Fm), non-photochemical quenching (NPQ) and effective quantum yield of photosystem II (Y (II)) in *L. aequinoctialis*.

Fv/Fm
HgCl_2_ (mg/L)	Day 0	Day 3	Day 5	Day 7
0	0.773 ± 0.018 ^aA^	0.760 ± 0.025 ^aA^	0.770 ± 0.019 ^aA^	0.779 ± 0.035 ^aA^
0.1	0.780 ± 0.015 ^aA^	0.772 ± 0.033 ^aA^	0.726 ± 0.203 ^abA^	0.745 ± 0.060 ^abA^
0.2	0.779 ± 0.023 ^aA^	0.743 ± 0.085 ^aA^	0.737 ± 0.044 ^abA^	0.733 ± 0.046 ^bA^
0.3	0.773 ± 0.011 ^aA^	0.640 ± 0.102 ^bB^	nd	nd

**NPQ**
**HgCl_2_ (mg/L)**	**Day 0**	**Day 3**	**Day 5**	**Day 7**
0	0.836 ± 0.633 ^aA^	0.858 ± 0.386 ^aA^	0.717 ± 0.540 ^aA^	0.827 ± 0.229 ^aA^
0.1	0.756 ± 0.404 ^aA^	1.457 ± 0.458 ^bB^	1.761 ± 0.421 ^bBC^	2.030 ± 0.290 ^bC^
0.2	0.770 ± 0.306 ^aA^	1.686 ± 0.545 ^bB^	2.019 ± 0.448 ^bBC^	2.241 ± 0.503 ^bC^
0.3	0.883 ± 0.262 ^aA^	3.712 ± 0.976 ^cB^	3.776 ± 0.839 ^cB^	3.692 ± 0.671 ^cB^

**Y (II)**
**HgCl_2_ (mg/L)**	**Day 0**	**Day 3**	**Day 5**	**Day 7**
0	0.191 ± 0.043 ^aA^	0.172 ± 0.028 ^aA^	0.220 ± 0.054 ^aA^	0.177 ± 0.044 ^aA^
0.1	0.191 ± 0.035 ^aA^	0.150 ± 0.019 ^bB^	0.204 ± 0.072 ^aA^	0.169 ± 0.017 ^abA^
0.2	0.184 ± 0.060 ^aA^	0.138 ± 0.033 ^bB^	0.140 ± 0.010 ^bB^	0.146 ± 0.025 ^bB^
0.3	0.210 ± 0.030 ^aA^	nd	nd	nd

Values represent mean ± SD. Different lowercase letters (a–c) within columns indicate significant differences among treatments (*p* < 0.05). Different uppercase letters (A–C) within rows indicate significant differences across days (*p* < 0.05). nd = tissue necrosis cannot be measured.

**Table 4 plants-14-02859-t004:** Effects of different concentrations of mercury on the maximum quantum efficiency of PSII photochemistry (Fv/Fm), the non-photochemical quenching (NPQ) and the effective quantum yield of photosystem II (Y (II)) in *S. polyrhiza*.

Fv/Fm
HgCl_2_ (mg/L)	Day 0	Day 3	Day 5	Day 7
0	0.779 ± 0.021 ^aA^	0.786 ± 0.025 ^aAB^	0.802 ± 0.019 ^aB^	0.803 ± 0.017 ^aB^
0.1	0.785 ± 0.022 ^aA^	0.780 ± 0.046 ^aA^	0.773 ± 0.041 ^aA^	0.772 ± 0.037 ^aA^
0.2	0.787 ± 0.013 ^aA^	0.712 ± 0.035 ^bAB^	0.601 ± 0.079 ^bBC^	0.500 ± 0.302 ^bC^
0.3	0.795 ± 0.020 ^aA^	0.694 ± 0.020 ^bA^	0.383 ± 0.288 ^cB^	0.336 ± 0.287 ^bB^

**NPQ**
**HgCl_2_ (mg/L)**	**Day 0**	**Day 3**	**Day 5**	**Day 7**
0	0.318 ± 0.165 ^aA^	0.326 ± 0.182 ^aA^	0.448 ± 0.163 ^aA^	0.442 ± 0.098 ^aA^
0.1	0.323 ± 0.204 ^aA^	0.364 ± 0.178 ^aA^	0.516 ± 0.223 ^aAB^	0.688 ± 0.295 ^bB^
0.2	0.319 ± 0.140 ^aA^	0.424 ± 0.343 ^aA^	0.985 ± 0.408 ^bB^	1.060 ± 0.285 ^cB^
0.3	0.322 ± 0.068 ^aA^	0.432 ± 0.098 ^aA^	1.230 ± 0.544 ^bB^	1.359 ± 0.375 ^dB^

**Y (II)**
**HgCl_2_ (mg/L)**	**Day 0**	**Day 3**	**Day 5**	**Day 7**
0	0.336 ± 0.066 ^aA^	0.344 ± 0.032 ^aA^	0.353 ± 0.033 ^aA^	0.351 ± 0.045 ^aA^
0.1	0.345 ± 0.050 ^aA^	0.337 ± 0.062 ^abA^	0.282 ± 0.030 ^bB^	0.274 ± 0.027 ^bB^
0.2	0.329 ± 0.058 ^aA^	0.272 ± 0.089 ^bcA^	0.176 ± 0.118 ^cB^	0.086 ± 0.039 ^cC^
0.3	0.332 ± 0.044 ^aA^	0.243 ± 0.108 ^cB^	0.098 ± 0.106 ^dC^	0.043 ± 0.027 ^dC^

Values represent mean ± SD. Different lowercase letters (a–c) within columns indicate significant differences among treatments (*p* < 0.05). Different uppercase letters (A–C) within rows indicate significant differences across days (*p* < 0.05).

## Data Availability

The data presented in this study are available on request from the corresponding author.

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
