# Peer review of "Comparative Physiological Responses of Lemna aequinoctialis and Spirodela polyrhiza to Mercury Stress: Implications for Biomonitoring and Phytoremediation"

_plants, 2025, doi:10.3390/plants14182859_

Round 1
Reviewer 1 Report
Comments and Suggestions for Authors
This study systematically compared the physiological responses of two types of floating duckweed (Lemna aequinoctilis and Spirodela polyrhiza) to mercury stress, and found that L. aequinoctilis is suitable as a biological monitoring species for mercury pollution, while S. polyrhiza has greater potential as a mercury remediation plant. The study provides species selection criteria and physiological theoretical basis for mercury pollution control in Thai water bodies. However, there are still some details worth paying attention to. If the author makes corresponding revisions and provides reasonable explanations, it can greatly increase the readability of the manuscript:
- On page 3, line 99, it is stated that the leaf area of S. polyrhiza decreased by 59.86% at 0.3 mg/L HgCl₂. However, Figure 1B shows no overlapping error bars between the control group (0 mg/L) and the 0.3 mg/L group (p<0.05), and the original data needs to be verified or the textual description needs to be corrected.
- The RFN line trend of S. polyrhiza in Figure 2 on page 4 does not match the text "similar downward trend" (the decrease in the figure is always lower than that of L. aequinoctalis), and additional statistical test results (such as letter annotations) are needed.
- On page 10, line 288 of the discussion section, it is mentioned that "Figure 6" displays chlorophyll content, but the entire text only has Figure 4 (A-D) and no Figure 6 was found. Unified chart numbering is required (such as Figure 4C-D corresponding to S. polyrhiza pigment data).
- The unit of "Carotenoid (mg/g (FW))" on the vertical axis of Figure 4D on page 6 is suspected to be incorrect (with a value as high as 54 mg/g). It is recommended to review whether the unit is "μg/g.
- The term "nd" (undetected) in Table 3 on page 7 should be clearly defined (such as "tissue necrosis cannot be measured").
- On page 12, line 393, the specific formula for the 10% Hoagland medium is not specified (does it contain chelating agent EDTA? May affect the bioavailability of mercury).
- In chlorophyll fluorescence measurement, the "ambient light conditions" need to specify the light intensity (μmol photons·m⁻²·s⁻¹), otherwise it is difficult to reproduce the experiment.
- On page 9, line 265 of the discussion section, it was mentioned that "mercury-induced colony breakup" was not correlated with the experimental results (time gradient in Table 1-2). Suggested supplement: Is the disintegration rate of L. aequinoctalis (1h vs S. polyrhiza 24h) related to differential expression of ethylene synthesis genes?
- The conclusion mentions the use of duckweed for water remediation in Thailand, but does not discuss the impact of environmental factors such as pH and coexisting ions on mercury absorption. Suggest supplementing short-term field experiment data or citing localized studies (such as mercury speciation analysis in the Mun River Basin of Thailand [8]) to eliminate limitations of research results.
- The introduction emphasizes the scarcity of mercury research on Thai duckweed, but does not highlight the newly discovered species in this study (L. aequinoctilis is endemic to Southeast Asia). Suggest adding the regional uniqueness of this species in the abstract and introduction paragraphs.
Author Response
1. On page 3, line 99, it is stated that the leaf area of polyrhiza decreased by 59.86% at 0.3 mg/L HgCl₂. However, Figure 1B shows no overlapping error bars between the control group (0 mg/L) and the 0.3 mg/L group (p<0.05), and the original data needs to be verified or the textual description needs to be corrected.
Response 1: Fig. 1B showed reduction of frond area (cm²) at 0.3 mg/L Hg treatment compared to the non-treatment group. The mean value of D7 non-treatment was 30.38 ± 0.78. The non-treatment group showed quite a bit of uniformity; thus, the SD bar was quite narrow. The mean value of the D7 0.3 mg/L Hg treatment was 12.20 ± 10.68. The D7 0.3 mg/L Hg treatment showed a high value in a biological replication; thus, the SD bar was a bit larger than the previous group.
2. The RFN line trend of polyrhiza in Figure 2 on page 4 does not match the text "similar downward trend" (the decrease in the figure is always lower than that of L. aequinoctalis), and additional statistical test results (such as letter annotations) are needed.
Response 1: Both Spirodela and Lemna showed similar declines. For Spirodela, RFN was 0.313 and then went down to 0.150, 0.100, and 0.066. For Lemna, RFN was 0.230, 0.087, 0.048, and 0.026. Statistical annotations appeared above the lines.
3. On page 10, line 288 of the discussion section, it is mentioned that "Figure 6" displays chlorophyll content, but the entire text only has Figure 4 (A-D) and no Figure 6 was found. Unified chart numbering is required (such as Figure 4C-D corresponding to polyrhiza pigment data).
Response 3: Thank you so much, we have fixed the errors. (on page 9, line 272)
4. The unit of "Carotenoid (mg/g (FW))" on the vertical axis of Figure 4D on page 6 is suspected to be incorrect (with a value as high as 54 mg/g). It is recommended to review whether the unit is "μg/g.
Response 4: Thank you. We have recalculated and fixed the figure. (on page 6, line 165)
5. The term "nd" (undetected) in Table 3 on page 7 should be clearly defined (such as "tissue necrosis cannot be measured").
Response 5: Thank you. The description was changed as suggested. (on page 7, line 197)
6. On page 12, line 393, the specific formula for the 10% Hoagland medium is not specified (does it contain chelating agent EDTA? May affect the bioavailability of mercury).
Response 6: We appreciate the reviewer’s comment regarding the potential effect of EDTA in 10% Hoagland solution on Hg bioavailability. In our experimental setup, EDTA is present as Fe-EDTA (ferric ethylenediaminetetraacetate), not as free EDTA. At the pH of Hoagland medium (~5.8–6.0), Fe(III) forms a highly stable complex with EDTA (log K ≈ 25.1), which is stronger than the Hg(II)–EDTA stability constant reported in similar conditions (log K ≈ 21.8). Thus, the Fe–EDTA complex is thermodynamically favored, and free EDTA concentration in the medium is negligible.
Furthermore, at the low EDTA concentration corresponding to 10% Hoagland strength, the molar amount of EDTA is just sufficient to maintain Fe solubility and is unlikely to sequester additional metals to a significant extent. Therefore, EDTA in our medium is not expected to appreciably alter Hg speciation or reduce its bioavailability to duckweed in our experiments.
We will keep in mind about this matter.
7. In chlorophyll fluorescence measurement, the "ambient light conditions" need to specify the light intensity (μmol photons·m⁻²·s⁻¹), otherwise it is difficult to reproduce the experiment.
Response 7: Thank you for suggestion. We added PPFD and temperature. (on page 10, line 334)
8. On page 9, line 265 of the discussion section, it was mentioned that "mercury-induced colony breakup" was not correlated with the experimental results (time gradient in Table 1-2). Suggested supplement: Is the disintegration rate of aequinoctalis (1h vs S. polyrhiza 24h) related to differential expression of ethylene synthesis genes?
Response 8: We appreciate the comment. We added one more reference mentioning external ethylene application, which resulted in decreasing frond per number of colonies in Lemna minor (on page 9, line 252). Thus, the expression of the ethylene synthesis gene could differ between L. aequinoctialis and S. polyrhiza.
9. The conclusion mentions the use of duckweed for water remediation in Thailand, but does not discuss the impact of environmental factors such as pH and coexisting ions on mercury absorption. Suggest supplementing short-term field experiment data or citing localized studies (such as mercury speciation analysis in the Mun River Basin of Thailand [8]) to eliminate limitations of research results.
Response 9: We added meta-analysis research that mentioned that pH affects Hg absorption. (on page 12, line 396).
10. The introduction emphasizes the scarcity of mercury research on Thai duckweed, but does not highlight the newly discovered species in this study (aequinoctilis is endemic to Southeast Asia). Suggest adding the regional uniqueness of this species in the abstract and introduction paragraphs.
Response 10: We added more description. (on page 2, line 64)
Reviewer 2 Report
Comments and Suggestions for Authors
All the equipment details, brand and model need to be provided in the section 4. The reliability of the Frond area calculation method, authors may provide the reference on the method applied. The software to apply ANOVA analysis need to be provided.
Authors need to explore the difference of Lemna aequinoctialis and Spirodela polyrhiza structure/ properties that causes difference performance of mercury absorption. Is there any difference phyoremediation mechanism involved in both Lemna aequinoctialis and Spirodela polyrhiza ? The mechanism of phytoremediation of mercury by the plants need to be explored and discussed.
Is the proposed method to remove mercury a green and sustainable approach? The plant will die after phytoremediation? What do to with the phytoremediated mercury plants? Will it be the secondary pollutants generation?
Author Response
1. All the equipment details, brand and model need to be provided in the section 4.
Response 1: Thank you for the comment. We added the details already.
2. The reliability of the Frond area calculation method, authors may provide the reference on the method applied.
Response 2: Reference added.
3. The software to apply ANOVA analysis need to be provided.
Response 3: Reference added.
4. Authors need to explore the difference of Lemna aequinoctialis and Spirodela polyrhiza structure/ properties that causes difference performance of mercury absorption. Is there any difference phyoremediation mechanism involved in bothLemna aequinoctialis and Spirodela polyrhiza ? The mechanism of phytoremediation of mercury by the plants need to be explored and discussed.
Response 4: Thank you for the suggestion. Research in duckweeds nowadays has not deeply studied the mechanisms of phytoremediation. We would say that the mechanism might be similar to other plants. More discussions were added on page 10, line 317.
5. Is the proposed method to remove mercury a green and sustainable approach? The plant will die after phytoremediation? What do to with the phytoremediated mercury plants? Will it be the secondary pollutants generation?
Response 5: Yes, that is another step to consider and find the solution to effectively remove the duckweeds.
Reviewer 3 Report
Comments and Suggestions for Authors
The authors have investigated Hg impact on Lemna aequinoctialis and Spirodela polyrhiza physiology.
Abstract contains unnecessary information (first three sentences). The last sentence also could not be proved by the results of the study.
In the Introduction a lot of attention was dedicated to Thailand, though no information on HG levels in the environment, especially aquatic ecosystems, was provided. Also, the usage of Lemna spp for phytoremediation must be justified and some recent studies referred. As Hg accumulation was not measured, the phytoremediation potential is speculative.
Results and Discussion: fresh and dry weight must be presented for 1 frond, not for the whole treatment.
Although two species were tested though no species effect was analysed, in addition the same is for time (in case for fluorescence measurements at different time points). The Discussion is well written but needs more justification with the additional measurements.
The main drawback of the study is that there is no linkage between measured morphological, and physiological endpoints with Hg accumulation as it was referred as the study of phytoremediation The study needs additional measurements of Hg accumulation and other parameters. But this was not done, Therefore, I suggest to resubmit improved manuscript.
Comments on the Quality of English Languagesome improvements are needed
Author Response
1. Abstract contains unnecessary information (first three sentences). The last sentence also could not be proved by the results of the study.
Response 1: Based on this study, Lemna are more sensitive to mercury than Spirodela; thus, Lemna can be used as biomonitors. Spirodela are more tolerant to mercury, and other research found that several duckweed species have the ability to be used as phytoremediators and can absorb mercury effectively. Additional texts were added in the discussion section (on page 10, line 322).
2. In the Introduction a lot of attention was dedicated to Thailand, though no information on HG levels in the environment, especially aquatic ecosystems, was provided.
Response 2: More information has been added (on page 2, line 45)
3. Also, the usage of Lemna spp for phytoremediation must be justified and some recent studies referred. As Hg accumulation was not measured, the phytoremediation potential is speculative.
Response 3: The answer has been shown in comment 1.
4. Results and Discussion: fresh and dry weight must be presented for 1 frond, not for the whole treatment.
Response 4: Thank you for the suggestion; however, duckweeds are tiny plants. Spirodela fronds are 5-10 mm long and 3-8 mm wide, and Lemna fronds are 1-6 mm long. Weighing a single frond would require a highly sensitive and precise balance, and we are concerned that this approach may lead to significant errors. Furthermore, other research weighed the whole treatment at once. For example, 10.1007/s10646-009-0408-0, 10.3390/plants12183206, and 10.1007/s11356-024-33583-5.
5. Although two species were tested though no species effect was analysed, in addition the same is for time (in case for fluorescence measurements at different time points). The Discussion is well written but needs more justification with the additional measurements.
Response 5: We set up tests for chlorophyll fluorescence at several time points: D0, D3, D5, and D7, using various concentrations. We also analyzed that L. aequinoctialis is more sensitive to mercury than S. polyrhiza, as shown on page 10, line 311.
6. The main drawback of the study is that there is no linkage between measured morphological, and physiological endpoints with Hg accumulation as it was referred as the study of phytoremediation The study needs additional measurements of Hg accumulation and other parameters. But this was not done.
Response 6: I appreciate the comment. The objective of this study was to observe different physiological responses of these two duckweed species. It is well known that duckweeds have potential as phytoremediation and accumulate metals, including Hg (the additional information was put in the discussion (on page 9, line 240)). The conclusion was drawn that Lemna cannot tolerate higher mercury concentrations; thus, it is appropriate for biomonitoring purposes. Spirodela are more tolerant, so they are better for use as phytoremediators. We also agree that measuring Hg accumulation would be additional information. (on page 11, line 402)
7. Comments on the Quality of English Language: some improvements are needed
Response 7: English Language was revised by the university service. Please see the attachment.

Round 2
Reviewer 3 Report
Comments and Suggestions for Authors
In most cases, the authors did not take into account the reviewer's comments and suggestions. The answers did not cover full the questions. therefore, main flawbacks of the study remained unsolved - no Hg accumulation, detailed analysis of plant weigh changes (of one frond, not whole colony and this does not require measurements), statistical analysis and analysis of the linkages between different endpoints. Therefore, I suggest to reject.
Author Response
1. Abstract contains unnecessary information (first three sentences). The last sentence also could not be proved by the results of the study.
Response 1: Thank you for the suggestion. We have already fixed it (on page 1, line 14 and line 26).
2. Results and Discussion: fresh and dry weight must be presented for 1 frond, not for the whole treatment.
Response 2: Thank you for the suggestion. We have recalculated. (on page 3, line 16)